# Kinematic Gait Analysis in People with Mild-Disability Multiple Sclerosis Using Statistical Parametric Mapping: A Cross-Sectional Study

**DOI:** 10.3390/s23187671

**Published:** 2023-09-05

**Authors:** Diego Fernández-Vázquez, Gabriela Calvo-Malón, Francisco Molina-Rueda, Raúl López-González, María Carratalá-Tejada, Víctor Navarro-López, Juan Carlos Miangolarra-Page

**Affiliations:** 1Physical Therapy, Occupational Therapy, Rehabilitation and Physical Medicine Department, Faculty of Health Sciences, Rey Juan Carlos University, 28922 Madrid, Spain; diego.fernandez@urjc.es (D.F.-V.); francisco.molina@urjc.es (F.M.-R.); maria.carratala@urjc.es (M.C.-T.); juan.miangolarra@urjc.es (J.C.M.-P.); 2Movement Analysis, Biomechanics, Ergonomics, and Motor Control Laboratory, Faculty of Health Sciences, Rey Juan Carlos University, 28922 Madrid, Spain; gonzalez.raul@urjc.es; 3School of Official Master’s Degrees, Universidad Rey Juan Carlos, 28922 Madrid, Spain; gabicalvomalon@gmail.com; 4Physical Medicine and Rehabilitation Department, University Hospital of Fuenlabrada, 28942 Madrid, Spain

**Keywords:** gait, multiple sclerosis, statistical parametric mapping

## Abstract

Multiple sclerosis (MS) is a chronic autoimmune disease that affects the central nervous system. Gait abnormalities, such as altered joint kinematics, are common in people with MS (pwMS). Traditional clinical gait assessments may not detect subtle kinematic alterations, but advances in motion capture technology and analysis methods, such as statistical parametric mapping (SPM), offer more detailed assessments. The aim of this study was to compare the lower-limb joint kinematics during gait between pwMS and healthy controls using SPM analysis. Methods: A cross-sectional study was conducted involving pwMS and healthy controls. A three-dimensional motion capture system was used to obtain the kinematic parameters of the more affected lower limb (MALL) and less affected lower limb (LALL), which were compared using the SPM analysis. Results: The study included 10 pwMS with mild disability (EDSS ≤ 3) and 10 healthy controls. The results showed no differences in spatiotemporal parameters. However, significant differences were observed in the kinematics of the lower-limb joints using SPM. In pwMS, compared to healthy controls, there was a higher anterior pelvis tilt (MALL, *p* = 0.047), reduced pelvis elevation (MALL, *p* = 0.024; LALL, *p* = 0.044), reduced pelvis descent (MALL, *p* = 0.033; LALL, *p* = 0.022), reduced hip extension during pre-swing (MALL, *p* = 0.049), increased hip flexion during terminal swing (MALL, *p* = 0.046), reduced knee flexion (MALL, *p* = 0.04; LALL, *p* < 0.001), and reduced range of motion in ankle plantarflexion (MALL, *p* = 0.048). **Conclusions**: pwMS with mild disability exhibit specific kinematic abnormalities during gait. SPM analysis can detect alterations in the kinematic parameters of gait in pwMS with mild disability.

## 1. Introduction

People diagnosed with multiple sclerosis (MS) often experience a continuous and progressive deterioration of their central nervous system, often affecting their mobility, bladder control, and cognitive abilities [1,2]. Research has shown that even people with mild disability due to MS have abnormalities in their gait compared to healthy people, as observed using three-dimensional motion capture systems. Specifically, these subjects walked with a reduced speed, stride length and cadence compared with the controls [3].

These gait abnormalities are associated with an increased risk of falls among people with MS [4]. In addition, several studies have reported that people with MS with mild disability have kinematic asymmetries, particularly in terms of reduced hip extension and hip range of motion during the stance phase [5,6]. These observations are even more pronounced in MS patients with spasticity [7]. In the ankle joint, some authors have observed decreased ankle dorsiflexion during initial contact and reduced plantarflexion during the pre-swing phase [8]. In addition, another work showed that mildly affected pwMS have altered muscle coactivation patterns during gait, especially in the most affected limb [9]. Interestingly, these alterations have not been adequately detected through non-instrumented performance-based tests used in clinical settings [3,10].

A previous study performed in pwMS with mild disability did not observe differences in kinetic and kinematic patterns, but a delay was observed in the temporal parameters of the range-of-motion gait cycle (ROM) [11]. Temporal parameters correspond to temporal occurrences, expressed as a percentage of the gait cycle, and to kinematic and kinetic parameters. For example, the peaks of hip extension and knee flexion during normal gait occur at approximately 50% and 70% of the completion of the gait cycle, respectively [12,13]. Specialized motion analysis systems are required to observe these differences.

Movement analysis plays a crucial role in the assessment of motor impairments associated with MS, as well as in understanding the functional limitations experienced by these individuals. In addition, movement analysis is useful in the development of personalized rehabilitation programs that address the specific movement deficits of each person and also in monitoring the evolution of the disease. Movement analysis in people with MS involves a comprehensive assessment of their gait, balance and coordination. Researchers use sophisticated technologies, such as motion capture systems, force plates and wearable sensors, to capture and quantify various movement parameters, comparing representative points, such as the degrees of hip extension at foot-off. However, statistical parametric mapping (SPM), originally developed to analyze cerebral blood flow, has begun to be used in biomechanics to compare the entire joint kinematic curve between two or more groups of subjects [14,15,16]. In a previous study, changes were observed in pwMS in the ankle, knee and hip joints compared with healthy controls. However, that study analyzed people with different forms of MS and moderate disability (a mean of 5.5 points on the Expanded Disability Status Scale (EDSS)) [17].

Based on our knowledge, there are no studies using SPM to analyze the kinematics of pwMS with mild disability (≤3 EDSS). Therefore, the aim of this study was to compare the lower-limb joint kinematics during gait between pwMS with ≤3 EDSS and healthy subjects using SPM.

## 2. Materials and Methods

### 2.1. Design

We conducted a cross-sectional study by asking for the voluntary participation of pwMS and healthy controls. The study was approved by the Research Ethics Committee of the Rey Juan Carlos University (registration code: 1202202006020), and written informed consent was obtained from all participants. The study was conducted according to the STROBE checklist [18].

### 2.2. Participants

Written informed consent was obtained before starting the evaluation. All procedures were carried out in accordance with the principles of the Declaration of Helsinki and the Biomedical Research Act 14/2007. Both pwMS and control subjects were asked to participate on a voluntary basis, with recruitment beginning in September 2022 and ending in January 2023. Patients had to meet the following inclusion criteria: (1) age > 18 years; (2) diagnosed with relapsing-remitting MS based on the 2017 McDonald criteria [19]; (3) EDSS score ≤ 3 [20]; and (4) no other comorbidities (e.g., musculoskeletal, cardiorespiratory, rheumatic, or other neurological diseases than MS). Participants who had in the previous six months experienced a worsening of symptoms, required hospitalization, received corticosteroid therapy (intravenous or oral), received botulinum toxin treatment, or experienced any other situation that could potentially hamper their participation in the study, were excluded. Healthy controls were recruited among family contacts in the pwMS environment. Recruited controls were matched for age and sex. Inclusion criteria for controls were: (1) age > 18 years; (2) ability to walk independently without assistance; and (3) absence of musculoskeletal and/or neurological disorders.

### 2.3. Experimental Protocol

The gait instrumental analysis was recorded by the Vicon Motion System^®^ (Vicon Motion Systems, Oxford, UK) using 8 MX 13+ infrared capture cameras, which were positioned so as to surround an 11 m walking corridor. Passive and reflective markers were placed over the skin, in specific anatomical standardized areas on the pelvis and lower limbs. These landmarks included the anterior and posterior superior iliac spines, middle third of the thigh, external femoral condyle, middle third of the tibia, external malleolus, calcaneus, and head of the second metatarsal, according to the biomechanical models established by Kadaba et al. [21] and Davis et al. [22]. Once the instrumentation was in place, the participants were instructed to walk at a self-selected comfortable gait speed along the corridor. The laboratory recorded ten repetitions per subject during each session. The Vicon^®^ Nexus software v2.13 was utilized to calculate outcome measures based on the biomechanical model of the Vicon^®^ Plug-in Gait. To minimize potential bias, the evaluator responsible for processing the gait tests worked independently from the researcher analyzing the results. The same protocol was followed for both study groups. All the measurements were performed at the Movement Analysis, Biomechanics, Ergonomics and Motor Control Laboratory (LAMBECOM) located at the Faculty of Health Sciences (Rey Juan Carlos University).

### 2.4. Outcome Measures

The spatiotemporal gait parameters and the kinematics (joint angles) were obtained from the average of 10 complete gait cycles for each of the patients. For the pwMS, data from both, the more affected lower limb (MALL) and the less affected lower limb (LALL), were used, obtaining a total of 100 gait cycles for each leg. For controls, 10 complete gait cycles were obtained as well, and subsequently a randomization process was performed in order to have an equal representation of the dominant and non-dominant leg, obtaining a total of 100 cycles (50 for each leg). Regarding the spatiotemporal parameters of gait, we analyzed the gait speed (m/s), stride length (m), step width (m), cadence (steps/min) and the timing of foot-off during the gait cycle (expressed as a percentage). The kinematic parameters were analyzed in 2 planes of space, sagittal and frontal, for the pelvis and hip, and in the sagittal plane for the knee and ankle.

### 2.5. Data Analysis

Spatiotemporal parameters were compared between pwMS and controls using a parametric mean comparison test, while gait kinematics were compared using the multivariate SPMHotelling t2 test, followed by univariate SPM t test comparisons for each joint. For each test, SPM{t2} or SPM{t} statistics were calculated, along with the threshold for rejection of the null hypothesis with a false-positive error rate of 0.05. The family-wise error rate of the SPM{t2} and SPM{t} tests was calculated for each joint. The error rate per family of the SPM *t*-tests was kept below 0.05 by the Holm–Bonferroni procedure. Comparisons were considered statistically significant if the test statistics exceeded the rejection threshold at one or more points along the continuum. In this case, points above the threshold formed one or more clusters, whose *p* values were calculated and reported. The analysis was performed using Python 3.9.7, Numpy (Python Software Foundation) and spm1d [23,24,25].

## 3. Results

The study involved a total of 10 persons with MS and 10 healthy subjects who participated in the research (Table 1). No patients had a diagnosis of the progressive type of MS. No differences in spatiotemporal parameters were observed between pwMS and healthy controls (Table 1).

### 3.1. Pelvis Kinematics

Significant findings were observed that indicated a higher anterior tilt during pre-swing (50.7–59.0% GC, *p* = 0.047) (Figure 1A) in MALL in pwMS. The frontal plane showed a reduced range of motion for both lower limbs. During the loading response and midstance phases, pwMS demonstrated less pelvic elevation (4.0–19.1% GC in the MALL, *p* = 0.024; and 7.9–14.5% GC in the LALL, *p* = 0.044) (Figure 1C,D). Additionally, during the pre-swing and initial swing phases, pwMS showed less pelvic descent (55.2–66.5% GC in the MALL, *p* = 0.033; and 51.8–68.8% GC in the LALL, *p* = 0.022) (Figure 1C,D).

### 3.2. Hip Kinematics

Significantly more hip flexion in terminal swing (93.9–100% GC; *p* = 0.046) and heel strike (0–1.5% GC; *p* = 0.049) and less hip extension in pre-swing (51.6–53.2% GC; *p* = 0.049) were observed in the MALL of the pwMS (Figure 2A). A lower adduction in the loading response (9.5–10.8% GC; *p* = 0.049) and a lower abduction in pre-swing (51.8–63.2% GC; *p* = 0.031) were observed in the MALL of the pwMS (Figure 2C). Regarding the LALL, in the frontal plane, less abduction between terminal support and initial swing (40.7–68.7% of the gait cycle; *p* = 0.012) was observed in pwMS (Figure 2D).

### 3.3. Knee Kinematics

Significantly reduced flexion during initial swing (61.6–66.5% GC, *p* = 0.040) was observed in MALL in pwMS (Figure 3A). There was also greater extension during mid-stance and terminal stance and reduced flexion during swing (30.8–71.5% GC, *p* < 0.001) in LALL in pwMS (Figure 3B).

### 3.4. Ankle Kinematics

Significantly greater dorsal flexion in terminal stance (46.8–47.1% GC; *p* = 0.049) and lower plantar flexion in pre-swing (56.5–58.1% GC; *p* = 0.048) were observed in MALL of pwMS (Figure 4A). Lower dorsal flexion in mid and terminal stance (27.6–43.6% GC; *p* = 0.022) and higher plantar flexion in initial swing (66.2–71.6% GC; *p* = 0.046) were observed in LALL of pwMS (Figure 4B).

## 4. Discussion

In this observational study, no differences have been observed between pwMS with an EDSS ≤ 3 and healthy subjects in the spatiotemporal parameters. However, significant differences have been found using SPM in the kinematics of the lower limb joints.

In our study, we observed an anterior tilt of the pelvis in the sagittal plane during pre-swing and a reduced range of motion in the frontal plane, similar to the findings of Mañago et al. [26]. They also observed this reduced range of motion in the frontal plane in 20 pwMS with an EDSS score ranging from 1.5 to 5.5. This restriction in the pelvic range of motion could be a compensatory mechanism to maintain pelvic stabilization.

Regarding the hip joint, a reduced extension during pre-swing and an increased flexion during terminal swing have been observed. This increased hip flexion during the terminal swing may be related to the increased anterior tilt of the pelvis in pwMS. According to the study by Benedetti et al. [27], the gait pattern of pwMS with mild disability is characterized by an increased hip flexion and reduced extension during pre-swing. These findings were also observed by Pau et al. [28] in individuals with MS with an EDSS between 2.5 and 4.5. In our study, we also observed a reduction in the hip range of motion in the frontal plane of the pwMS, consistent with the findings of a study conducted by Severini et al. [29], which observed a reduced frontal plane hip motion and increased sagittal plane pelvis motion in 52 individuals with MS (EDSS 2–6.5) during the swing and push-off phases of walking. These studies suggest that restricting frontal plane pelvic motion while walking could hold functional significance for people with MS. It might be a potential strategy to improve stability and ultimately enhance overall balance. However, in the study by Mestanza et al. [17], these hip differences were not observed in individuals with MS using SPM, possibly because fewer gait cycles were used than in our study and healthy subjects were asked to walk at a slower speed to match pwMS. These differences in methodology between our study and the study involving healthy subjects may have contributed to the variations in results. By using fewer gait cycles and altering the gait speed of healthy subjects, the natural walking mechanics and patterns might have been altered. As a result, the comparison between the two groups may not have accurately reflected the genuine differences in gait between healthy individuals and pwMS.

In the knee joint, a reduction in flexion during the initial swing phase was observed. In the study by Filli et al. [30], a similar reduction in knee flexion was observed in pwMS with a mean EDSS of 4.5. This reduction in flexion is considered a predictor of gait functionality in pwMS [25]. In our case, we observed this knee change in pwMS with a lower EDSS score of less than 3.0, suggesting that this knee impairment may be present in early stages of the disease and in patients with mild disability. In the study by Mestanza et al. [17] using SPM, similar changes were observed in pwMS, with a reduction in flexion between 68–76% of GC. In our study, this reduction was observed between 61.6–66.5% for the MALL and between 30.8–71.5% for the LALL. In the study by Pau et al. [28], which investigated the effect of lower-limb spasticity on kinematics by examining both kinematics and muscle activation, a significant reduction in knee flexion and extension was found in people with MS who had higher levels of spasticity, which could be related to the increased activation of the rectus femoris muscle observed in this group. This reduction in knee flexion in the initial oscillation may also be due to a lower pre-oscillatory impulse.

Regarding the ankle joint, a significantly reduced range of motion in dorsiflexion in midstance and terminal stance and a reduced plantarflexion in pre-swing were observed in pwMS. This reduction in the ankle range of motion may be due to an increased coactivation of the tibialis anterior and triceps surae as a compensatory strategy to achieve greater joint stability [31]. The reduction in plantarflexion at pre-swing and initial swing has been observed in several studies [28,32] and is related to this compensatory mechanism, which is aimed at increasing stability to reduce falls. This reduction can also be attributed to an increased muscle tone, weakness of the triceps surae and reduced hip flexor strength [32]. The reduced pre-oscillatory impulse by the triceps surae associated to a decrease in hip extension in this phase may be related to a shorter stride length, even though no alterations in the spatiotemporal parameters were observed in the present study, possibly because the alterations were subtle due to the low level of disability of pwMS. As for the reduction in dorsiflexion, in our study we observed between 27.6–43.6% of the GC in the LALL, which is similar to the findings of Mestanza et al. [17], who observed a significant reduction in dorsiflexion between 20–49% of the gait cycle.

In the study by Morel et al. [33], the Gait Profile Score tool was used to assess the kinematics of pwMS with an EDSS ≤ 2. They found significant differences compared to the healthy control group in the ankle and pelvis, but not in the knee and hip. This difference, when compared to our study (where we did find differences in the knee and hip joints), may be attributed to the fact that the SPM analysis may have a higher sensitivity in detecting these alterations in pwMS with mild disability. Therefore, SPM may be a more appropriate tool for kinematic analysis in the early stages of the disease. These findings highlight the presence of altered gait patterns in individuals with MS, even in the early stages of mildly disabled disease. The use of SPM can contribute to the assessment and management of people with MS, particularly in the identification of possible compensatory mechanisms. In the study conducted by Galea et al. [34], a deterioration of gait and balance was observed in pwMS that was not reflected in the clinical status measured with the EDSS. This highlights the importance of laboratory-based assessments analyzed with sensitive techniques, such as SPM, to determine the motor control deterioration in each joint in pwMS.

This study has some clinical implications. Evaluation methods such as SPM can be more sensitive in detecting changes in joint kinematics, enabling a more accurate and early assessment of motor function. This can guide the development of specific rehabilitation interventions to improve gait and stability in individuals with MS, consequently enhancing their independence and quality of life. Additionally, tools such as SPM could also inform targeted interventions in the field of rehabilitation or the development of new technologies for MS management.

There are several limitations to this study. First, the sample size of pwMS was relatively small, so caution should be exercised in generalizing the results. Therefore, future studies with larger sample sizes are needed. Secondly, our study did not include lower-limb electromyography data, which limits the interpretation of our results. In future studies, the inclusion of muscle activity during the gait cycle may be valuable to relate it to the kinematic changes found.

## 5. Conclusions

SPM analysis shows that pwMS with mild disability show changes in their kinematic gait pattern. According to this analysis, pwMS walk with a higher anterior pelvis tilt, reduced pelvis range of motion in the frontal plane, reduced hip extension during pre-swing, increased hip flexion during terminal swing, reduced knee flexion during the initial swing phase, and reduced range of motion in ankle dorsiflexion and plantarflexion. These findings are mainly observed in the most affected lower limb.

## Figures and Tables

**Figure 1 sensors-23-07671-f001:**
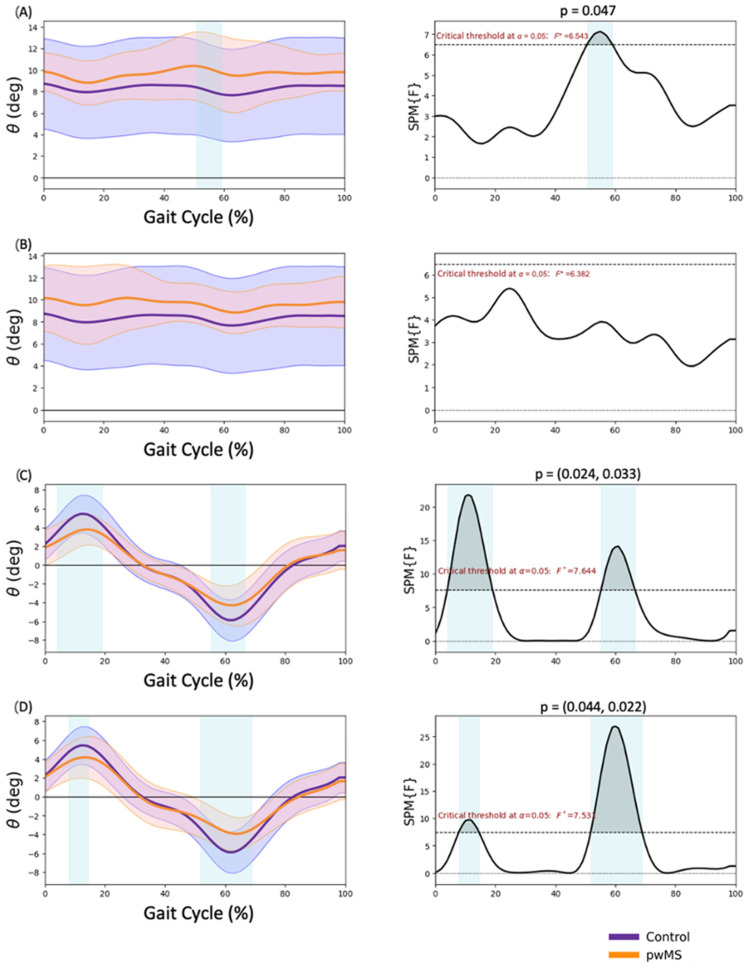
Pelvis kinematics. (**A**) More affected lower limb sagittal plane, (**B**) less affected lower limb sagittal plane, (**C**) more affected lower limb frontal plane, (**D**) less affected lower limb frontal plane. Positive values are pelvis anterior tilt in the sagittal plane and pelvis lift in the frontal plane. Negative values are pelvis posterior tilt in the sagittal plane and pelvis drop in the frontal plane. * *p* < 0.05.

**Figure 2 sensors-23-07671-f002:**
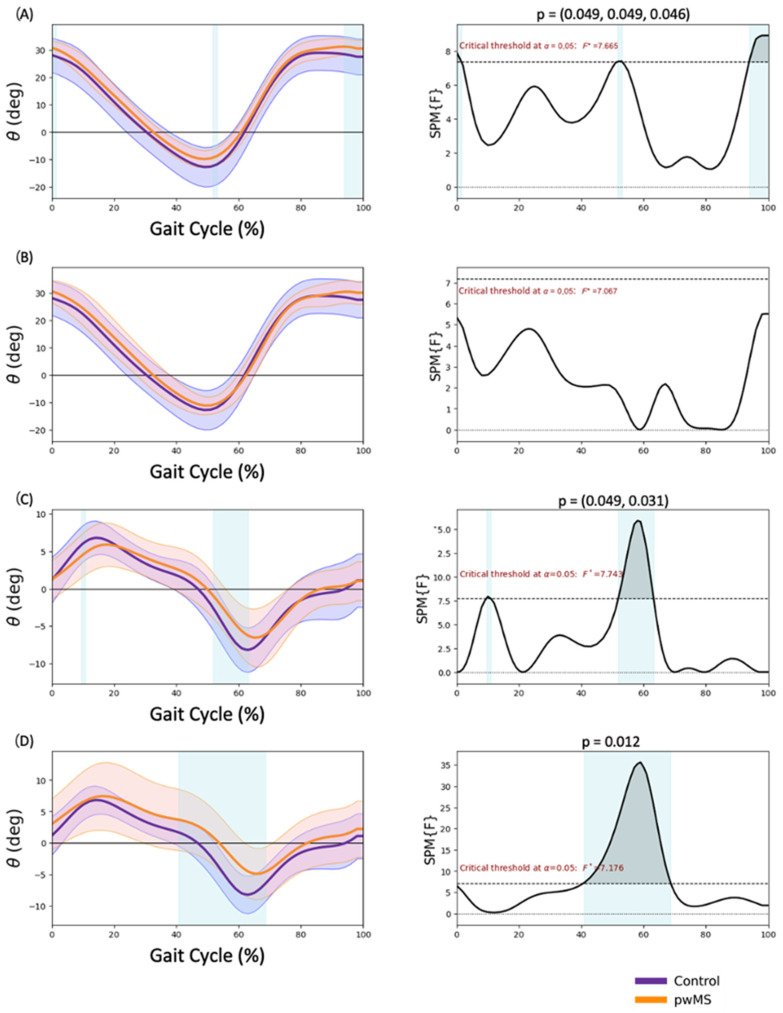
Hip kinematics. (**A**) More affected lower limb sagittal plane, (**B**) less affected lower limb sagittal plane, (**C**) more affected lower limb frontal plane, (**D**) less affected lower limb frontal plane. Positive values are hip flexion in the sagittal plane and hip adduction in the frontal plane. Negative values are hip extension in the sagittal plane and hip abduction in the frontal plane. * *p* < 0.05.

**Figure 3 sensors-23-07671-f003:**
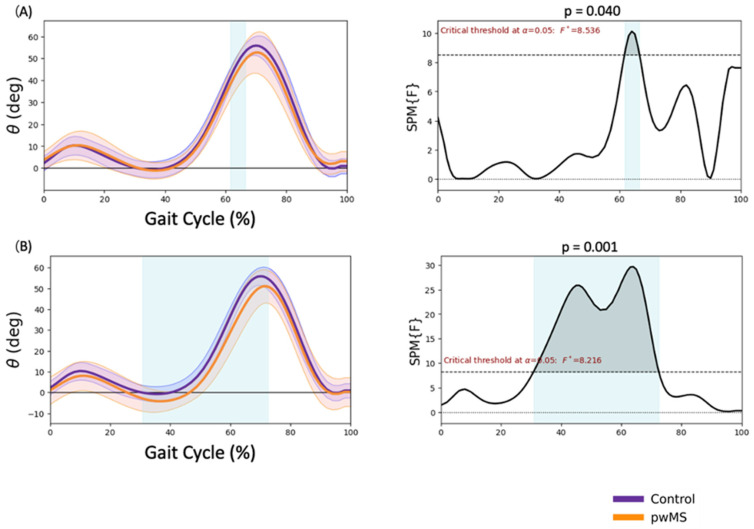
Knee kinematics. (**A**) More affected lower limb sagittal plane, (**B**) less affected lower limb frontal plane. Positive values are knee flexion. Negative values are knee extension. * *p* < 0.05.

**Figure 4 sensors-23-07671-f004:**
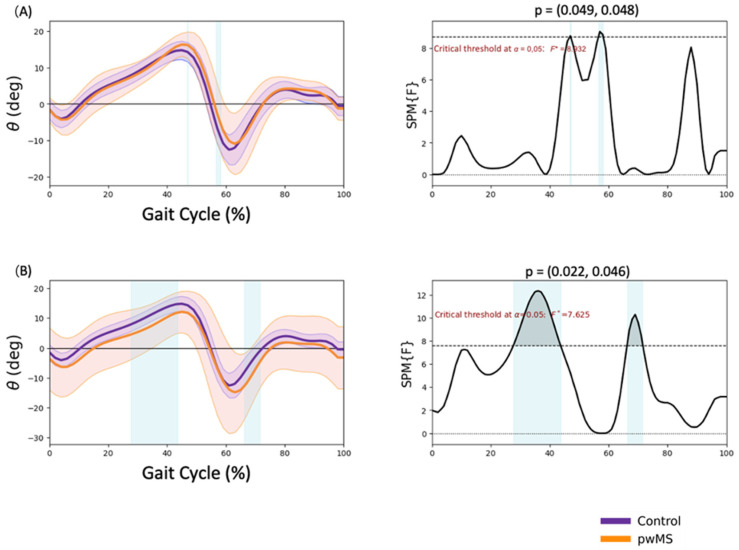
Ankle kinematics. (**A**) More affected lower limb sagittal plane, (**B**) less affected lower limb frontal plane. Positive values are ankle dorsal flexion. Negative values are plantar flexion. * *p* < 0.05.

**Table 1 sensors-23-07671-t001:** Demographic data and spatiotemporal gait parameters.

Parameters	Multiple Sclerosis	Controls	*p*
Gender	7 female/3 male	7 female/3 male	
Age	35.8 (9.1)	35.7 (9.3)	0.981
Height (cm)	1.7 (0.08)	1.73 (0.07)	0.432
Weight (kg)	68.6 (8.6)	66.2 (14.1)	0.65
Years since diagnosis	9.12 (8.82)		
EDSS (Median IQR)	2.25 (1.9)		
Gait speed (m/s)	1.16 (0.19)	1.24 (0.1)	0.098
Stride length (m)	1.27 (0.13)	1.3 (0.08)	0.248
Steep width (m)	0.09 (0.02)	0.07 (0.03)	0.105
Cadence (m)	110.91 (12.03)	113.8 (6.65)	0.353
Foot-off (%)	61.12 (2.57)	60.58 (1.76)	0.441

EDSS—Expanded Disability Status Scale, IQR—interquartile range.

## Data Availability

No application.

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
