# Peer review of "Kinematic Gait Analysis in People with Mild-Disability Multiple Sclerosis Using Statistical Parametric Mapping: A Cross-Sectional Study"

_sensors, 2023, doi:10.3390/s23187671_

Round 1

Reviewer 1 Report

The study aimed to use SPM analysis to compare kinematic parameters of the lower limbs (MALL & LALL) in individuals with MS with low EDSS during walking versus kinematic parameters  in age and sex match healthy controls.

The study was well conducted and clearly presented. All the figures are clear.

The information from this study incrementally advances the understanding of altered gait in people with MS at early stage, which would help clinicians in assessing gait of people with MS during rehabilitation process.

There are some minor typos need to be improved.

1/ Inclusion criteria # 4: should be "NO other comorbidities..."

2/ Please rephrase line 129, page 3

3/ Line 284, page 10: remove "was" 

Author Response

Thank you very much for the revision process, the authors consider that with the changes, the article has improved with respect to the previous version.

Reviewer 1

There are some minor typos need to be improved.

1/ Inclusion criteria # 4: should be "NO other comorbidities..."

Thank you for your comments, inclusion criteria 4 has been changed.

2/ Please rephrase line 129, page 3

We have rewrite line 129

3/ Line 284, page 10: remove "was"

We have remove was in line 284

Reviewer 2 Report

This research presents original results concerning the comparison between the human lower limb  kinematics during gait between pwMS and healthy controls using SPM analysis and significant differences were observed. A cross-sectional study was conducted involving pwMS and healthy controls.

The paper is very interesting and relevant for the research field of effects of MS on the gait parameters. The paper is well written, correctly organized, with appropriate title, abstract, introduction, notations, content, conclusions and references. The current stage is exhaustive, critical, with presentations of the approaches of other authors, the results obtained by them, with clear references to the specialized literature published in this field. But, we recommend an improvement of the state of the art with 2-3 more cited articles.

The Discussions and Conclusions chapter presents very clearly the results and comparative analyzes with important conclusions The figures, tables and mathematical expressions in the paper are correct from a scientific and technical point of view. 

The paper is correctly written in international English. 

Author Response

Reviewer 2

“But, we recommend an improvement of the state of the art with 2-3 more cited articles.”

Thank you for your comments, we have included more cited articles.

Reviewer 3 Report

Reviewer comments

Fernandez-Vazquez et al. conducted a study that sought to identify any differences in gait between individuals with multiple sclerosis (MS) and controls, in which the individuals with MS had relatively low disability scores as determined using the EDSS. They recorded movements using an infrared motion capture/tracking system, and extracted a series of spatiotemporal gait features. They additionally used statistical parametric mapping (SPM) to compare trajectories of gait cycles in individuals with MS and controls. They found that, although spatiotemporal features failed to capture differences between groups, the SPM-based method was able to capture distinctions in the MS group. These included various changes in movement patterns in the pelvis, hips, and ankles.

Overall, I find this paper to be well-written and interesting. It is appropriately concise, and implements a novel use for SPM to good effect. I have a few minor suggestions to improve this article, itemized below, after which I would recommend it for publication.

1.     At lines 94-95: “presence of other comorbidities…” etc. is listed as (4) in the inclusion criteria, but I think this is meant to instead be listed as exclusion criteria?

2.     At lines 125-126: Why did the controls do 100 gait cycles total instead of 100 per leg with randomization (i.e., 200 cycles total)?

3.     Line 129: “cycle in when occurs the foot off (%).” This wording is slightly awkward, I would recommend rephrasing.

English is generally good. See comments for details

Author Response

Reviewer 3

  1. At lines 94-95: “presence of other comorbidities…” etc. is listed as (4) in the inclusion criteria, but I think this is meant to instead be listed as exclusion criteria?

Thank you for your comment, we have changed the way the inclusion criteria 4 was written.

  1. At lines 125-126: Why did the controls do 100 gait cycles total instead of 100 per leg with randomization (i.e., 200 cycles total)?

We have collected 100 gait cycles per leg from the healthy controls. However, as there are no significant differences between both legs (non-dominant and dominant), to ensure an equivalent dataset to that of the persons with MS, we have randomly chosen 50 cycles from both the dominant and non-dominant legs for comparison with the 100 cycles of each leg from persons with MS. Since we believe it is important for SPM to perform the comparison with an equal number of cycles in both groups.

We have included in the manuscript that 10 gait cycles were obtained from the controls for improved clarification.

  1. Line 129: “cycle in when occurs the foot off (%).” This wording is slightly awkward, I would recommend rephrasing.

Thank you, we have rewrite this phrase.